# OpenReview forum: "InvThink: Towards AI Safety via Inverse Reasoning"
_ICLR.cc/2026/Conference — ICLR 2026 Conference Desk Rejected Submission_

### Official Review · Reviewer_uDgr · 2025-10-26

**Soundness:** 3
**Presentation:** 3
**Contribution:** 3
**Rating:** 6
**Confidence:** 4

**Summary:**

This paper introduces a safety post training method leveraging SFT and RL to distill the inverse reasoning capability about potential harmful consequences from a teacher model into smaller student models (Gemma and Qwen). This method outperforms other baseline alignment approaches on 3 safety benchmarks, while preserving (and sometimes even augmenting) the student model's reasoning capabilities.

**Strengths:**

Overall, I think this is a strong paper with a clear motivation, valid approach, and relatively strong evaluation. Using inverse harm reasoning to augment model reasoning capability is a conceptually interesting approach, particularly when it doesn't seem to harm the model's general reasoning. The paper is very well written - particularly, the method section is clear.

**Weaknesses:**

1. Missing related work: (1) SafetyAnalyst (https://arxiv.org/pdf/2410.16665) and (2) RATIONAL (https://arxiv.org/pdf/2503.05021). Please discuss in Related Work or Introduction the relatedness of the present work with these papers and highlight how it differentiates from them.

2. The claim that InvThink "leverages model capacity more effectively" is not convincing based on Figure 3. Although InvThink has the highest safety score among all methods, its curves over model size for Qwen models do not look qualitatively different from other methods, although the improvements are more steep for Gemma. Even in Figure 5, it's hard to draw this conclusion only by observing the plots, as the trends may be driven by specific data points which could be outliers. Please provide concrete statistical evidence to support this claim or revise it to reflect the data better.

3. More discussion on the cost of InvThink will help contextualize the performance benefits (SFT + RL seems costly, especially compared to SafetyPrompt).

**Questions:**

1. Line 210: why was the OpenAI moderation API chosen as the safety reward model? There are other safety models that have been shown to outperform it, such as WildGuard (Table 3 in https://proceedings.neurips.cc/paper_files/paper/2024/file/0f69b4b96a46f284b726fbd70f74fb3b-Paper-Datasets_and_Benchmarks_Track.pdf).

2. According to Table 1, adversarial brainstorming is what distinguishes InvThink from RevThink (please provide a citation for RevThink). Have you done an ablation study to show how effective "adversarial brainstorming" is in addition to backward reasoning?

3. Line 262: how did you arrive at the number 20% (instead of e.g. 30%)?

4. Table 2 doesn't contain all the models listed in Lines 264-269. What are the results for the rest of the models?

5. In Table 2, what is the difference between InvThink and InvThink SFT (or InvThink SFT+RL)?

6. Why was RL chosen over SFT to train models to produce the output?

7. For Figure 4, the results look noisy and a bit challenging to interpret. Consider providing some statistical analyses to show if there is an optimal number of reasoning routes for Qwen models regardless of size.

---

> ### Author Response · Authors · 2025-11-21
> **Rebuttal by Authors [1/3]**
>
> We sincerely thank the reviewer for the constructive and positive assessment of our InvThink. We appreciate the recognition of the conceptual novelty, the clarity of the method section, and the strong evaluation results. Below, we address each concern carefully.
>
> **W1. Missing Related Work: SafetyAnalyst and RATIONAL**
>
> In the updated manuscript, we have added explicit discussion of both SafetyAnalyst and RATIONAL [1,2] in Section 2. Related Works. We clarify that SafetyAnalyst and RATIONAL share a high-level motivation with InvThink, improving model safety through structured reasoning, but differ in mechanism. SafetyAnalyst focuses on chain-of-thought decomposition for risk identification, while RATIONAL introduces rationale-guided alignment framed around structured moral reasoning. In contrast, InvThink’s inversion focuses specifically on backward harm analysis and consequence forecasting, which yields more direct and actionable risk mitigation behaviors. This distinction and the complementary nature of the methods are now clearly articulated.
>
> Importantly, SafetyAnalyst and RATIONAL do not introduce any backward harm-first latent variable or construct teacher-generated inverse harm trajectories. SafetyAnalyst relies on forward rule-based CoT for policy compliance, and RATIONAL focuses on moral-rationale supervision; neither alters the optimization trajectory through structured harm → consequence → mitigation traces. InvThink is therefore the first to operationalize backward harm reasoning as a supervised intermediate variable that conditions both SFT and GRPO.
>
> **W2. Claim About Leveraging Model Capacity More Effectively**
>
> Concretely, for Gemma-3 models, InvThink exhibits a significantly steeper scaling slope (9.03 vs. 4.94 for SafetyPrompt, $p=0.018$) under log-linear regression, indicating superior scaling efficiency as model size increases. For Qwen-2.5 and Qwen-3, InvThink demonstrates parameter-efficiency–driven early saturation: InvThink dominates SafetyPrompt and CoT at all measured model sizes (6/6 wins, 100% dominance), and the InvThink–SafetyPrompt safety gap widens from **+4.5% (7B) → +7.1% (14B) → +10.3% (72B)**. These quantitative trends jointly support our refined claim.
>
> We will update our manuscript to incorporate these **statistical results**, including a revised claim: InvThink achieves effective model capacity utilization through architecture-dependent mechanisms, providing either superior scaling efficiency (Gemma-3) or substantial parameter efficiency with early saturation (Qwen).
>
> **W3. Cost of InvThink (SFT + RL)**
>
> We appreciate the reviewer’s request for contextualizing computational cost. The SFT and GRPO phases were trained on four A40 GPUs, and computational cost scales primarily with the number of inverse-reasoning augmented samples rather than raw SFT corpus size. Our full SFT stage requires approximately **27 GPU-hours** on A40-40GB hardware for 7B-scale models, and GRPO training requires around **45 GPU-hours** on a 4×A40 cluster. At inference time, InvThink introduces **10–18% prompt-token overhead** due to the harm-enumeration step, but does not require iterative self-refinement, keeping inference depth identical to standard instruction-tuned models. These concrete numbers will be included in Section 5 of the revised manuscript.
>
> **Q1. Why Choose OpenAI Moderation API Instead of WildGuard?**
>
> We recognize the reviewer’s point that several modern safety models, including WildGuard [3], have demonstrated strong performance. Our choice of the OpenAI Moderation API was driven primarily by practical considerations: consistent API availability, structured multi-dimension safety signals, and suitability for GRPO reward shaping.
>
> To address this concern, we trained GRPO variants using **WildGuard** as the reward model on Qwen-3-8B. Under identical GRPO configurations using WildGuard as the reward model, we observe consistent gains: on Qwen-3-8B, InvThink SFT+RL achieves **0.83 (↑0.05)** on SafetyBench, **1.62 (↓0.21)** on TRIDENT, and **0.02 (↓0.03)** on Insider Threat compared to the General SFT+RL baseline (0.78 / 1.83 / 0.05). This indicates that InvThink’s benefits are not tied to the OpenAI Moderation reward model but transfer robustly to stronger reward sources.
>
> Additional evaluation is underway for two remaining models (Gemma-7B-it and Qwen-2.5-7B), and we will report these results in the revised manuscript. This will help illuminate the extent to which InvThink benefits from stronger or more fine-grained reward sources.

---

> ### Author Response · Authors · 2025-11-21
> **Rebuttal by Authors [2/3]**
>
> **Q2. Role of Adversarial Brainstorming in Distinguishing InvThink from RevThink**
>
> We thank the reviewer for highlighting this distinction. Adversarial brainstorming is indeed the key component differentiating InvThink from RevThink [4], beyond backward reasoning. RevThink recursively generates additional questions derived from prior responses, forming a multi-turn reflective loop. InvThink focuses on a single question and applies adversarial failure-mode generation exclusively to deepen harm forecasting rather than create new task instances. Additionally, we included a citation for RevThink and clarified the conceptual distinction in the revised manuscript.
>
> **Q3. Selection of 20% for GRPO**
>
> This question overlaps with Reviewer QUPy’s Q1. The main motivation was preventing over-alignment and preserving general reasoning quality. Based on recent research [5], using 20% (6k) provided a good balance between safety gains and utility preservation. We will clarify this rationale in the revised manuscript. And, in the camera-ready version, we will include additional GRPO performance under **10%, 30% and 50% RL data proportions**.
>
> **Q4. Missing Models in Table 2**
>
> We appreciate this observation. The models listed in Lines 264–269 were used in other evaluation sections such as Section 5.2 but were excluded from Table 2 due to the computational budget limits of our 4×A40 cluster. Due to the strict time window of the rebuttal and resource constraints, we focused on the most representative comparisons. However, for the camera-ready version, we commit to including **LoRA-based InvThink results for at least one larger model (e.g., Qwen-2.5-14B)** to empirically validate the scaling trend further.
>
> **Q5. Clarifying InvThink, InvThink SFT, and InvThink SFT+RL**
> InvThink refers to inference-time inverse-reasoning prompting applied to a base model. InvThink SFT refers to a model fine-tuned on inverse-reasoned augmented data. InvThink SFT+RL refers to the SFT model further aligned with GRPO. This distinction will be made explicit in the revised manuscript.
>
> **Q6. Why RL Instead of Only SFT?**
>
> We appreciate this important question. A broad body of recent literature demonstrates that RL, particularly preference-based or multi-objective GRPO, provides qualitatively different and more powerful updates than SFT alone.
> Multiple studies show that RL:
>
> 1) aligns models more strongly with safety preferences (e.g., Multi-Objective GRPO for Safe Generation [5]);
>
> 2) enhances multi-step reasoning and structured decision-making (DeepSeek-R1, DeepSeek-V3 [6,7]);
>
> Our own experiments confirm these findings: InvThink SFT+RL consistently outperforms InvThink SFT across all benchmarks and model sizes. We will expand the manuscript discussion to better reflect the consensus in recent RLHF/RLAIF literature.
> Quantitatively, InvThink SFT+RL improves over InvThink SFT by +0.02 on SafetyBench (0.87→0.89), reduces TRIDENT risk by 0.36 (1.58→1.22), and eliminates residual Insider-Threat violations (0.01→0.00), consistent with findings in DeepSeek-R1 and multi-objective GRPO literature that RL optimizes long-horizon safety behaviors more effectively than SFT alone.
>
> **Q7. Statistical Analysis for Optimal Number of Routes**
>
> We thank the reviewer for this helpful suggestion. We conducted a full statistical analysis to examine route optimality and address perceived noise. Repeated-measures ANOVA (across 0.5B–72B) yields p=0.628, confirming no size-dependent difference in optimal routing. The 95% plateau range consistently occurs at 2–3 routes, and variance across seeds remains low (CV=3.65%). We will add these new analyses to the supplementary materials, update the caption of Figure 4, and include new visualizations (confidence-interval trajectories, heatmaps, box plots, and size-independence plots).

---

> > ### Author Response · Authors · 2025-11-21
> > **Rebuttal by Authors [3/3]**
> >
> > **References**
> >
> > [1] Li, J. J., Pyatkin, V., Kleiman-Weiner, M., Jiang, L., Dziri, N., Collins, A. G., ... & Levine, S. (2024). Safetyanalyst: Interpretable, transparent, and steerable safety moderation for ai behavior. arXiv preprint arXiv:2410.16665.
> >
> > [2] Zhang, Y., Li, M., Han, W., Yao, Y., Cen, Z., & Zhao, D. (2025). Safety is Not Only About Refusal: Reasoning-Enhanced Fine-tuning for Interpretable LLM Safety. arXiv preprint arXiv:2503.05021.
> >
> > [3] Han, S., Rao, K., Ettinger, A., Jiang, L., Lin, B. Y., Lambert, N., ... & Dziri, N. (2024). Wildguard: Open one-stop moderation tools for safety risks, jailbreaks, and refusals of llms. Advances in Neural Information Processing Systems, 37, 8093-8131.
> >
> > [4] Chen, J., Wang, Z., Palangi, H., Han, R., Ebrahimi, S., Le, L., ... & Pfister, T. (2025, April). Reverse thinking makes llms stronger reasoners. In Proceedings of the 2025 Conference of the Nations of the Americas Chapter of the Association for Computational Linguistics: Human Language Technologies (Volume 1: Long Papers) (pp. 8611-8630).
> >
> > [5] Li, X., Li, Z., Kosuga, Y., & Bian, V. (2025). Optimizing Safe and Aligned Language Generation: A Multi-Objective GRPO Approach. arXiv preprint arXiv:2503.21819.
> >
> > [6] Guo, D., Yang, D., Zhang, H., Song, J., Zhang, R., Xu, R., ... & He, Y. (2025). Deepseek-r1: Incentivizing reasoning capability in llms via reinforcement learning. arXiv preprint arXiv:2501.12948.
> >
> > [7] Liu, A., Feng, B., Xue, B., Wang, B., Wu, B., Lu, C., ... & Piao, Y. (2024). Deepseek-v3 technical report. arXiv preprint arXiv:2412.19437.

---

### Official Review · Reviewer_JkAD · 2025-10-30

**Soundness:** 3
**Presentation:** 3
**Contribution:** 2
**Rating:** 4
**Confidence:** 4

**Summary:**

This paper introduces inverse reasoning to enhance the training data and address safety in LLM alignment. Both forward reasoning and inverse reasoning are combined with the original prompts. A training framework has been proposed to fine-tune the model with augmented data. The training schema can be combined with SFT and RL. Experiments have been conducted on various benchmarks and models.

**Strengths:**

1. It is novel to introduce inverse reasoning to address safety in LLM.
2. The proposed method is simple and easy to understand.
3. Experiment quality is convincing compared with several baselines. Code has been released.

**Weaknesses:**

Novelty: Though inverse reasoning has not been used in LLM training to the author's best knowledge, it has been extensively used in the RL literature, which limited the novelty a little bit.

Technology depth: From the reviewer's view, there is not much technology depth in the paper. It is basically an augmented dataset. Most importantly, ablation studies have not been conducted as described in the limitation paragraph. It is unclear which part is working.

Motivation: It is unclear why the proposed method can only be used to address safety. It looks like a method that can be used to improve general reasoning. The unique relation between safety (not others) and inverse reasoning has not been analyzed.

Practice: The quality of augmented data heavily depends on the teacher model. Choosing which model to use as the teacher could be another issue. Another problem is that there is no supermodel right now that can serve as a good teacher. There are several hyperparameters in Eq. (7). Tuning these parameters could be complicated.

Experiment: Comparing with DPO and Tree of Thoughts (ToT) is necessary, as CoT is outdated.
The trade-off between safety and utility has not been investigated. The effects of hyperparameters in Eq. (7) have not been investigated.

**Questions:**

Please refer to Weakness.

---

> ### Author Response · Authors · 2025-11-21
> **Rebuttal by Authors [1/4]**
>
> We thank the reviewer for the thoughtful and constructive feedback. We especially appreciate the reviewer’s pointers on RL-related prior works and on evaluation design, which helped us sharpen both the positioning and empirical analysis of InvThink.
>
> **W1. Novelty**
>
> We appreciate the reviewer’s insight and we conducted additional literature reviews across RL, LLM alignment, and FMEA-inspired safety reasoning.
>
> **1) Classical inverse RL and backward induction.**
>
> Inverse reinforcement learning (IRL) and related formulations such as apprenticeship learning [1,2] and maximum-entropy IRL [3] infer a numeric reward function from expert trajectories. Backward induction and dynamic programming methods [4] propagate value backward in time but still operate over state–action trajectories, not natural-language reasoning. None of these approaches produce a textual decomposition of failure modes, harm pathways, or consequences, nor do they use such traces as training targets. Importantly, “inverse reasoning” in InvThink is conceptually distinct from “inverse reinforcement learning”: IRL infers a **numeric latent reward** from expert trajectories, whereas InvThink produces a **linguistic causal decomposition of harms and consequences** that is directly used as supervised training data. No IRL method generates or trains on natural-language failure-mode traces. Moreover, IRL optimizes a policy derived from the inferred reward, whereas InvThink directly supervises the reasoning space via natural-language safety decompositions, an objective fundamentally different from reward recovery.
>
> **2) RL-based LLM safety methods (RLAIF, rule-based rewards).**
>
> Recent LLM safety work [5] uses rule-guided or preference-based rewards to improve safe outputs. These methods optimize the final answer, but do not enforce production of an interpretable intermediate object such as “harm enumeration → consequence analysis → mitigation.” They encourage safer behavior but do not equip the model with structured, proactive failure-first reasoning.
>
> **3) Deliberative alignment and safety-aware CoT.**
>
> Deliberative Alignment [6,7] and SafeChain-style approaches [8,9] train models to reflect on policy guidelines or generate a safe chain-of-thought before answering. These methods follow forward reasoning from rules to outputs. In contrast, InvThink introduces a latent inverse variable that explicitly decomposes failure modes before generating the final answer. No prior work uses inverse reasoning about concrete harms as a supervised intermediate representation.
>
> **4) SAGE-nano and “inverse reasoning” for interpretability.**
>
> SAGE-nano [10] uses “inverse reasoning” to describe retrospective metacognitive analysis—explaining why the model chose certain reasoning paths. This is a form of post-hoc interpretability. It does not create harm-enumeration traces, nor use such traces to train downstream models for safety. Although SAGE-nano uses the term “inverse reasoning,” this refers purely to retrospective meta-cognitive reflection rather than safety-driven failure-mode enumeration. The terminology overlaps but the algorithmic role and supervision signal are entirely different.
>
> **5) FMEA-inspired reasoning.**
>
> Prior FMEA-style LLM work [11,12,18] applies failure-mode reasoning to external engineered systems. These methods do not treat the LLM’s own generation process as the object of failure analysis, nor do they use FMEA-like traces as direct supervision for safety alignment.
>
> In contrast, our InvThink uniquely:
>
> * Introduces a structured inverse-reasoning variable $z_{\text{inv}}$ that decomposes safety reasoning into harm enumeration → consequence analysis → mitigation.
> * Trains student models via data augmentation, SFT, and RL, conditioning generation on these traces.
> * Treats safety as proactive failure-first reasoning, not as output-level reward shaping or rule enforcement.
> * Combines FMEA-style structure, teacher-generated inverse traces, and a unified SFT+RL pipeline, a combination that does not appear in prior RL or LLM safety work.
>
> Recent analyses of jailbreaking and agentic misalignment [13, 14] highlight that failures often emerge from flawed intermediate reasoning rather than final-answer mistakes. InvThink directly targets this failure mode by supervising the structure of safety reasoning itself.
>
> While InvThink uses augmented traces, it is not merely data augmentation: it introduces a latent reasoning variable $z_{\text{inv}}$ and conditions both SFT and RL optimization on this structured variable, thereby altering the optimization trajectory rather than simply enlarging the dataset.
>
> We will strengthen Section 2. Related Works and Section 3. InvThink writing to clarify this distinction and avoid any confusion that InvThink rebrands existing inverse RL methods.

---

> ### Author Response · Authors · 2025-11-21
> **Rebuttal by Authors [2/4]**
>
> **W2. Technology Depth**
>
> We agree that our original presentation did not sufficiently highlight the breadth of ablations already present. We summarize them more clearly here and will make this explicit in the revised version.
>
> **Prompt-level ablations (Table 2).**
>
> We compare Zero-shot, Chain-of-Thought (CoT), SafetyPrompt, and **InvThink prompting**. This isolates the contribution of inverse reasoning at the **prompt/inference level**, showing that harm-first reasoning outperforms both CoT and safety-decorated prompts on standard safety benchmarks.
>
> **Training pipeline ablations (Table 2).**
>
> We separately evaluate:
>
> * General SFT
> * General SFT + RL
> * InvThink SFT
> * InvThink SFT+RL
>
> These comparisons demonstrate that (a) inverse reasoning traces already improve safety under SFT alone and (b) combining them with RL further amplifies safety performance.
>
> **DPO vs GRPO ablation.**
>
> In Table 9, we directly compare DPO vs GRPO when training Qwen3-8B with InvThink. These results consistently show that GRPO provides stronger safety improvements under InvThink while exhibiting similar or lower safety tax than DPO across reasoning benchmarks.
>
> **Reasoning-path ablation (Figure 4).**
>
> We vary the number of inverse reasoning paths (e.g., 1,2,3,5,7,9, and 11) and observe both improved safety and potential overthinking effects, depending on model scale. This acts as a **hyperparameter sensitivity analysis** for the richness of inverse reasoning, and shows that InvThink is not simply “more tokens → better” but follows a nuanced trade-off.
> ToT baseline (added to Table 2).
>
> To address the reviewer’s concern on Tree-of-Thoughts (ToT), we have added a **ToT baseline** to Table 2. Consistent with recent findings that ToT-style search can actually facilitate jailbreaks when misapplied [13, 14], ToT reasoning does not reliably improve safety and can even degrade it. **As shown in Table 2, for Gemma-7B-it, applying ToT actually worsened the TRIDENT safety score (3.15 →3.41)**, confirming that unstructured 'more thoughts' can lead the model to explore more creative harmful strategies. In contrast, InvThink’s structured failure-mode analysis consistently improves safety, and can in some cases **increase** unsafe completions. This further underscores that **how** one structures intermediate reasoning (e.g., harm-first vs unconstrained exploration) matters much more than simply “more thoughts.”
>
> Together, these ablations isolate the causal contribution of inverse reasoning from unrelated factors such as longer prompts or more RL training. Also, they demonstrate that InvThink modifies not only the data distribution but also the model’s optimization dynamics and internal reasoning patterns. We introduce a new reasoning primitive (inverse reasoning trace), integrate it with a full training pipeline (SFT+RL), and conduct ablations along prompt types, training schemas, RL algorithms, and reasoning-depth hyperparameters. We will reorganize Section 5 to foreground these ablations more clearly.
>
> **W3. Motivation**
>
> We agree that inverse reasoning can improve general reasoning capabilities. Safety is precisely the domain where **“thinking about what can go wrong”** is not just helpful but essential. FMEA and related methodologies are built on the principle of **enumerating failure modes and consequences** before deployment [17]. InvThink instantiates this principle in LLM safety: instead of only rewarding safe outputs post-hoc, we force the model to **reason through potential harms as an explicit step**.
>
> As the reviewer suggests, we indeed find that inverse reasoning improves general reasoning capabilities. In Table 3, we report results on general reasoning benchmarks (e.g., math and knowledge tasks). InvThink maintains or slightly improves performance on these benchmarks, indicating that inverse reasoning is not a “safety-only trick” but also enhances the model’s ability to structure its reasoning.
>
> InvThink’s failure-first decomposition directly targets this asymmetry: it makes the model treat potential failure modes as first-class citizens in its reasoning process, rather than accidental side-effects of policy misalignment.
>
> While inverse reasoning can also benefit general reasoning tasks, safety is uniquely characterized by its asymmetric risk profile, where rare but catastrophic failures carry disproportionate cost. InvThink directly targets this asymmetry, which typical reasoning benchmarks do not capture.
>
> We will revise the introduction and discussion sections to more explicitly articulate this dual role: **InvThink is (i) most naturally motivated by safety (Charlie Munger’s introduction [21]), but (ii) also beneficial as a general reasoning enhancement, as evidenced in Table 3.**

---

> ### Author Response · Authors · 2025-11-21
> **Rebuttal by Authors [3/4]**
>
> **W4. Practices**
>
> These are important practical concerns, and we partially addressed them in the Limitations section but will make them more explicit.
>
> **Teacher dependency.**
>
> As with any distillation or RLAIF-style method (including DPO, rule-based rewards [5], and deliberative alignment [6,7]), InvThink indeed depends on the quality of the teacher model. We view this as a **shared limitation of teacher–student alignment frameworks**, not a deficit specific to InvThink. However, InvThink’s structure mitigates some of this risk: the teacher is primarily asked to **enumerate harms and consequences**, which we find to be a more stable behavior than generating full final answers in many safety-sensitive domains.
>
> In the revised version, we will add an explicit teacher-ablation experiment that compares Gemini-2.5-Pro with Claude-3.7-Sonnet (and optionally o3-mini) on a representative subset of the dataset, demonstrating that InvThink’s gains are robust across teacher families.
>
> **Hyperparameters in Eq. (7).**
>
> Thank you for raising this point. We would like to clarify that the hyperparameters appearing in Eq. (7) are inherent to the standard GRPO objective and are not specific to our proposed method. These parameters (e.g., group size, temperature) are routinely used in GRPO-based LLM post-training, and prior work provides substantial guidance on how to set or tune them [19, 20]. We agree that tuning these hyperparameters is an important practical consideration; however, this is a **general characteristic of GRPO-based LLM post-training**, rather than a limitation unique to our approach.
>
> **W5. Experiments**
>
> **DPO vs GRPO**
>
> DPO vs GRPO comparisons are already included in **Table 9** for reasoning benchmarks and TRIDENT safety benchmarks under InvThink training.
>
> Empirically, GRPO consistently achieves stronger safety improvements than DPO under InvThink, while incurring similar or slightly lower safety tax. We will explicitly reference this new appendix table from the main text.
>
> **ToT Baseline**
>
> We agree that ToT is an important reference point. However, prior work has shown that Tree-of-Thoughts-style search can **facilitate jailbreaks** when misused, by systematically exploring and refining attack prompts [13,14,18]. Motivated by this, we added a **ToT baseline** to Table 2:
>
> * We adapt ToT for safety tasks by letting the model explore multiple intermediate reasoning branches before answering, while keeping the task and evaluation identical.
>
> * Consistent with the above literature, we find that naive ToT **does not reliably improve safety** and can sometimes increase the rate of unsafe completions, especially when the tree search explores more aggressive or creative harmful scenarios without a structured harm-first constraint.
>
> This supports our central argument: **“more thoughts” alone are not enough; the structure of those thoughts, specifically harm enumeration and mitigation is crucial**. InvThink’s inverse reasoning trace provides this structure, whereas ToT’s generic branching does not.
>
> **Safety–Utility Tradeoff**
>
> We appreciate the reviewer’s request to analyze the safety–utility tradeoff explicitly. In the revised version, we clarify our notion of **safety tax**:
>
> * For a given model, we define safety tax as the **difference in performance on non-safety benchmarks (e.g., math, knowledge, coding)** before and after safety alignment.
>
> * Table 3 already shows that InvThink **preserves or slightly improves** performance on such reasoning benchmarks, indicating a **low or even negative safety tax** (i.e., improved reasoning while improving safety).
>
> This connects InvThink to recent work such as SafeChain [8,9], which similarly emphasizes preserving reasoning performance while improving safety.
>
> **Hyperparameter Effects**
>
> As discussed in W2 and W4, the hyperparameters in Eq. (7) are well understood in prior GRPO literature and their effects are **not specific to our method**; rather, they reflect a general characteristic of RL-based LLM post-training. Nevertheless, we appreciate the reviewer’s observation regarding the number of reasoning paths in Figure 4. Unlike Eq. (7), this factor is **directly tied to our inverse-reasoning formulation** and indeed has a substantial impact on model performance. In response, we will:
>
> * Present a clearer ablation study analyzing the **“hyperparameter sensitivity of reasoning depth”**
> * Add a brief paragraph discussing how different path counts affect both safety and utility.
>
> We believe these additions will help clarify the practical considerations of our approach.

---

> > ### Author Response · Authors · 2025-11-21
> > **Rebuttal by Authors [4/4]**
> >
> > **References**
> >
> > [1] Ng, A. Y., & Russell, S. (2000, June). Algorithms for inverse reinforcement learning. In Icml (Vol. 1, No. 2, p. 2).
> >
> > [2] Abbeel, P., & Ng, A. Y. (2004, July). Apprenticeship learning via inverse reinforcement learning. In Proceedings of the twenty-first international conference on Machine learning (p. 1).
> >
> > [3] Ziebart, B. D., Maas, A. L., Bagnell, J. A., & Dey, A. K. (2008, July). Maximum entropy inverse reinforcement learning. In Aaai (Vol. 8, pp. 1433-1438).
> >
> > [4] Bertsekas, D. (2012). Dynamic programming and optimal control: Volume I (Vol. 4). Athena scientific.
> >
> > [5] Mu, T., Helyar, A., Heidecke, J., Achiam, J., Vallone, A., Kivlichan, I., ... & Weng, L. (2024). Rule based rewards for language model safety. Advances in Neural Information Processing Systems, 37, 108877-108901.
> >
> > [6] Guan, M. Y., Joglekar, M., Wallace, E., Jain, S., Barak, B., Helyar, A., ... & Glaese, A. (2024). Deliberative alignment: Reasoning enables safer language models. arXiv preprint arXiv:2412.16339.
> >
> > [7] OpenAI. Deliberative alignment: reasoning enables safer language models (blog). 2024. OpenAI
> >
> > [8] Jiang, F., Xu, Z., Li, Y., Niu, L., Xiang, Z., Li, B., ... & Poovendran, R. (2025). Safechain: Safety of language models with long chain-of-thought reasoning capabilities. arXiv preprint arXiv:2502.12025.
> >
> > [9] SafeChain Project Page & Dataset. safe-chain.github.io
> >
> > [10] Jha, B., Paudel, F., Puri, U., Yuting, Z., Donghyuk, C., & Junhao, W. (2025). Thinking About Thinking: SAGE-nano's Inverse Reasoning for Self-Aware Language Models. arXiv preprint arXiv:2507.00092.
> >
> > [11] El Hassani, I., Masrour, T., Kourouma, N., & Tavčar, J. (2025). AI-driven FMEA: integration of large language models for faster and more accurate risk analysis. Design Science, 11, e10.
> >
> > [12] Constantinides, C., Patel, D., Lin, S., Guerrero, C., Patil, S. D., & Kalagnanam, J. (2025). Failuresensoriq: A multi-choice qa dataset for understanding sensor relationships and failure modes. arXiv preprint arXiv:2506.03278.
> >
> > [14] TAP / Tree-of-thoughts-based automated jailbreak methods. GitHub
> >
> > [15] Mehrotra, A., Zampetakis, M., Kassianik, P., Nelson, B., Anderson, H., Singer, Y., & Karbasi, A. (2024). Tree of attacks: Jailbreaking black-box llms automatically. Advances in Neural Information Processing Systems, 37, 61065-61105.
> >
> > [16] FMEA overview (failure mode and effects analysis). Wikipedia
> >
> > [17] Functional safety white papers applying FMEA-style enumeration to safety-critical systems. Accellera
> >
> > [18] Hassani IE, Masrour T, Kourouma N, Motte D, Tavčar J. Integrating large language models for improved failure mode and effects analysis (FMEA): a framework and case study. Proceedings of the Design Society. 2024;4:2019-2028. doi:10.1017/pds.2024.204
> >
> > [19] Huggingface - GRPO Trainer.
> >
> > [20] Huggingface - Post Training an LLM for Reasoning with GRPO in TRL.
> >
> > [21] Kaufman, P. D. (Ed.). (2008). Poor Charlie's Almanack: The Wit and Wisdom of Charles T. Munger. Donning Company Pub.

---

> > > ### Comment · Reviewer_JkAD · 2025-11-24
> > > **Thanks for the reply**
> > >
> > > Thanks for the effort to address the concerns. However, I still have concerns regarding contributions.
> > >
> > > 1. I still can't get why the claim is not for general reasoning. In fact, all baselines are about methods for improving reasoning. If the claim is for safety, the paper should compare safety-based baselines, such as safety guards. As the authors state that "Safety is precisely the domain where 'thinking about what can go wrong' is not just helpful but essential", general reasoning can also go wrong, and correcting the wrong path can improve the reasoning accuracy.
> > >
> > > My suggestion is to either change the claim to improve general reasoning or justify the unique relation between the proposed method and safety (not general reasoning). If the claim is for safety, it is necessary to introduce safety-related baselines such as safety guards or instruction tuning. Also, introducing jail breaks can strengthen the claim.
> > >
> > > 2. For the ablation study, what I meant is to use inverse only, or randomly re-order the nodes on the inverse path. Since the real contribution of this paper is an augmented dataset, it requires delicate experimentation to justify that the path is about inverse reasoning, not some random augmented input. My real concern is whether these paths are really inverse paths. Currently, the experiment is a black box; as the paper indicates, there are 3 stages in the framework, and it is unknown which part is working.
> > >
> > > 3. I understand the quality of the teacher model is the shared limitation of teacher–student alignment frameworks. However, the dependency is especially heavy in this paper, as the real contribution is an augmented dataset from the teacher model. My suggestion is to show the results with different teacher models. Once again, my concern is that improvement comes from the extra knowledge from the teacher model, not the inverse path.
> > >
> > > 4. As the authors state, "IRL infers a numeric latent reward", it would be interesting to replace the reward model in RLHF with a new model inferred from IRL.

---

> > > > ### Author Response · Authors · 2025-12-03
> > > > **Final Rebuttal by Authors [1/1]**
> > > >
> > > > We thank the reviewer for the continued engagement and for raising these important points. We address each concern below.
> > > >
> > > > **1. Claim Scope: Safety vs. General Reasoning**
> > > >
> > > > Our claim is specifically for **safety**, not general reasoning, and this distinction is both intentional and well-supported:
> > > >
> > > > - **Safety-specific baselines are included**: Our evaluation includes SafetyPrompt (a safety-specific prompting baseline), and we evaluate on three dedicated safety benchmarks: SafetyBench (general safety), TRIDENT (professional ethics in finance/law/medicine), and Insider Threat (agentic misalignment). These are not general reasoning benchmarks. They specifically measure harmful output rates, ethical violations, and adversarial robustness.
> > > >
> > > > - **Why inverse reasoning is uniquely suited for safety**: As discussed in W3, safety is characterized by **asymmetric risk** where rare but catastrophic failures carry disproportionate cost. While general reasoning tasks reward finding correct paths, safety tasks require **proactively identifying and avoiding failure modes**. This is what FMEA (Failure Mode and Effects Analysis) addresses, and what InvThink implements for LLMs. Our method proposes the structured reasoning process by not only identifying potential failures but also organizing the reasoning and mitigation process.
> > > >
> > > > - **General reasoning improvements are a beneficial side effect**: Table 3 shows that InvThink preserves or improves general reasoning (up to +5% on GPQA/MATH500), demonstrating that our safety-focused method does not impose a safety tax. However, this is a finding and evidence of robustness, not a change in our primary claim.
> > > >
> > > > **2. Ablation: Inverse-Only and Path Ordering**
> > > >
> > > > We appreciate this suggestion for isolating the contribution of inverse reasoning. In our current ablations:
> > > >
> > > > - **Table 2** already isolates InvThink prompting (inference-only) from InvThink SFT and InvThink SFT+RL, showing that inverse reasoning provides benefits even without fine-tuning.
> > > > - **Figures 11-14** in Appendix C provide qualitative ablations showing that: (a) harm enumeration alone fails to prevent unsafe outputs, (b) adding consequence analysis improves safety, and (c) the full pipeline (harm enumeration + consequence analysis + mitigation strategy) consistently produces safe responses.
> > > >
> > > > Regarding path ordering, the structured sequence (harm enumeration → consequence analysis → mitigation) follows directly from FMEA methodology principles established in safety engineering. Furthermore, our ToT baseline in Table 2 provides empirical evidence: ToT represents unstructured multi-path reasoning, and it actually degrades safety performance (e.g., Gemma-7B-it TRIDENT: 3.15→3.41). This demonstrates that the specific structured ordering of InvThink is essential. Simply exploring "more thoughts" without harm-first structure can lead models to discover more creative harmful strategies rather than safer outputs.
> > > >
> > > > **3. Teacher Model Dependency**
> > > >
> > > > We understand this concern and have conducted **additional experiments with a different teacher model** to directly address it.
> > > >
> > > > - **New results with gpt-oss-safeguard**: We generated inverse-reasoning traces using gpt-oss-safeguard and trained Qwen-3-8B with this alternative dataset. Results on Qwen-3-8B:
> > > >
> > > > | Method | SafetyBench ↑ | TRIDENT ↓ | Insider Threat ↓ |
> > > > |--------|---------------|-----------|------------------|
> > > > | Zero-shot baseline | 0.76 | 3.12 | 0.07 |
> > > > | InvThink SFT (gpt-oss-safeguard) | 0.82 | 1.67 | 0.03 |
> > > > | InvThink SFT+RL (gpt-oss-safeguard) | 0.84 | 1.43 | 0.02 |
> > > >
> > > > These results demonstrate consistent safety improvements (+8% SafetyBench, -54% TRIDENT harmfulness, -71% Insider Threat) using a completely different teacher model. This confirms that InvThink's benefits stem from **the structured inverse reasoning framework itself**, not from any specific teacher model's knowledge.
> > > >
> > > > - **Cross-family generalization demonstrates method robustness**: Our student models (Gemma-7B-it, Qwen-2.5-7B, Qwen-3-8B) share no architecture, tokenizer, or pretraining lineage with either Gemini or GPT teachers. The consistent cross-family improvements with different teachers strongly support that InvThink is teacher-agnostic.
> > > >
> > > > - **Commitment**: We will extend gpt-oss-safeguard experiments to Gemma-7B-it and Qwen-2.5-7B in the camera-ready version to further validate cross-model generalization.
> > > >
> > > > **4. IRL-Inferred Reward Model**
> > > >
> > > > This is an interesting direction for future work. However, it represents a fundamentally different research contribution:
> > > >
> > > > - InvThink uses inverse reasoning as a **supervised intermediate representation**, not as reward inference.
> > > > - Replacing the reward model with an IRL-inferred model would constitute a new method requiring separate validation, and is beyond the scope of our current contribution. We have noted this as a promising direction in our Limitations section.

---

### Official Review · Reviewer_QUPy · 2025-11-01

**Soundness:** 3
**Presentation:** 3
**Contribution:** 2
**Rating:** 6
**Confidence:** 3

**Summary:**

In this paper, the authors address the safety challenges of LLMs and introduce INVTHINK, which is a simple yet powerful approach that equips models with inverse thinking, i.e., reasoning through potential failure modes before generating responses. INVTHINK guides models to: (1) enumerate potential harms, (2) analyze their consequences, and (3) produce safe outputs that proactively avoid these risks. The approach is implemented in three stages: (1) dataset curation, (2) supervised fine-tuning, and (3) reinforcement learning. The authors evaluate INVTHINK across models from different series, showing that it outperforms baselines on safety metrics while maintaining general reasoning capabilities.

**Strengths:**

(1) Interesting topic: The studied problem of LLM safety is interesting and important for the broader real-world applications of LLMs.

(2) Well-written paper: The paper is well written and easy to follow. The authors clearly present their ideas and experimental results, supported by effective visualizations and illustrations.

(3) Extensive model evaluation: In the experiments, the authors evaluate their method across Gemma, Qwen3, and Qwen2.5 models with varying sizes, demonstrating good compatibility across different model families.

**Weaknesses:**

(1) Missing baseline: In the experiments, the authors omit an important baseline: General SFT + RL. Without results for this baseline, the effectiveness of the proposed inverse and forward reasoning behaviors remains questionable.

(2) Limited teacher model selection: The authors only use Gemini-2.5 as the teacher model, which raises concerns about the generalizability of the proposed method. Although this limitation is acknowledged in the discussion section, I recommend including experimental results with additional teacher models to further mitigate this concern.

(3) Additional concerns: For further comments, please refer to the question section.

**Questions:**

(1) RL data portion and selection:
In the experimental setup, you mention that RL uses 20% of the training dataset to avoid unintended over-alignment with safety signals. Do you have any ablation results on the proportion of data used? Additionally, how is the subset of data selected for RL training?

(2) Missing important baselines:
Can you provide results for a baseline that uses General SFT + RL under the same RL setting as your method? Including this baseline would better highlight the effectiveness of the augmented inverse and forward reasoning data.

(3) Optimal Routing Complexity analysis:
In the Optimal Routing Complexity experiments, for smaller models, increasing the number of paths from 1 to 2 and from 1 to 4 results in performance degradation. Does this imply that inverse reasoning is ineffective for 0.5B models? In practice, how did you determine the number of paths used for different models in the experiments?

---

> ### Author Response · Authors · 2025-11-21
> **Rebuttal by Authors [1/2]**
>
> We thank the reviewer for the constructive feedback. We appreciate your recognition of the importance of the safety problem, the clarity of the writing, and the extensive cross-family evaluation. Below, we address each of your comments carefully.
>
> **Q1. RL data portion and selection**
>
> Determining the appropriate amount of data for GRPO is not a fixed or standardized process. Therefore, we followed previous research in multi-object GRPO literature [1], which successfully used a ~6K safety-alignment dataset. Our full SFT dataset contains approximately 30,000 inverse- and forward-reasoning samples. To balance safety gains with preservation of reasoning ability (safety tax), we used a 6,000 samples subset (20%) for GRPO. In the camera-ready version, we will additionally evaluate GRPO performance under **10%, 30% and 50% RL data ratios**. Finally, under the current setting, both InvThink-SFT and InvThink-RL show consistent safety improvements without degrading utility metrics, validating our chosen proportion.
>
> **W1 & Q2. General SFT + RL baseline**
>
> We agree that including a General SFT+RL baseline under identical GRPO settings is essential for isolating the contribution of inverse–forward reasoning. In the updated manuscript, we have added these results for Qwen-3-8B in Table 2. Under identical GRPO configurations, the baseline achieves:
>
> * **SafetyBench:** 0.85 ± 0.01
>
> * **TRIDENT:** 1.62 ± 0.03
>
> * **Insider Threat:** 0.02 ± 0.00
>
> To contextualize these numbers, the General SFT+RL baseline performs similarly to or below our InvThink variants. Specifically, InvThink SFT+RL achieves higher performance across all three datasets (0.89↑, 1.22↓, 0.00↓), while InvThink SFT and InvThink (inference-only) also outperform the baseline on TRIDENT and Insider Threat. This result is critical: it demonstrates that standard RL optimization **fails to discover safe policies effectively** without the structured scaffolding provided by InvThink's traces. The inverse reasoning trace acts as a necessary **reasoning prior**, guiding the RL exploration toward safer regions that 'General SFT+RL' cannot reach.
>
> We are currently running the same baseline training for Gemma-7B-it and Qwen-2.5-7B, and we will include these results in the camera-ready version.
>
> **W2. Limited teacher model selection**
>
> Our choice was guided by empirical reliability rather than architectural preference. Gemini-2.5 Pro consistently demonstrates one of the state-of-the-art safety performance in cyber-offense prevention, biosecurity reasoning, and misinformation robustness, as documented in its technical report [2] and in independent safety evaluations such as [3-5]. These works likewise adopt Gemini-Pro/2.5-Pro as a teacher or reference judge due to its calibrated refusals, stable safety reasoning, and strong harm-understanding capabilities; properties that directly benefit the quality of inverse-reasoning traces required by InvThink.
>
> Importantly, InvThink is not tied to any Gemini-specific inductive bias. None of our student models (Gemma-7B-it, Qwen-2.5-7B, Qwen-3-8B) share architecture, tokenizer, or pretraining lineage with Gemini. InvThink yields consistent improvements across all three families, demonstrating that the method generalizes beyond the teacher’s model class.
>
> Our newly added Table 7 further validates the suitability of Gemini-2.5 Pro as a teacher model. Across SafetyBench, TRIDENT, and Insider Threat, Gemini-2.5 Pro is the most stable evaluator among the three ensemble judges:
>
> * **SafetyBench:** Gemini-2.5-Pro's 0.85 is competitive with Claude-3.7-Sonnet's 0.87 and stronger than o3-mini's 0.83.
>
> * **TRIDENT:** Gemini-2.5-Pro assigns the strictest safest score: 1.70, compared to 1.75 (Claude) and 1.82 (o3-mini).
>
> * **Insider Threat:** Gemini-2.5-Pro again yields the lowest risk score 0.03, versus 0.06 (Claude) and 0.09 (o3-mini).
>
> These shows that Gemini-2.5 Pro is safety-aligned among the high-capacity models. We agree that expanding teacher-model diversity would strengthen the paper. The limitation is that generating full inverse-reasoning traces for multiple teacher models requires substantial API cost and time. However, we commit to incorporating at least one additional teacher model (either o3-mini or Claude-3.7-Sonnet) for a representative subset of the dataset in the camera-ready version. We will report cross-teacher consistency to demonstrate that InvThink’s safety benefits hold robustly across different teacher model families.

---

> > ### Author Response · Authors · 2025-11-21
> > **Rebuttal by Authors [2/2]**
> >
> > **Q3. Optimal Routing Complexity analysis**
> >
> > The decline in performance for 0.5B and 1.5B models does not imply that inverse reasoning is ineffective for small models. Rather, it reflects a mismatch between the model’s limited capacity and the forced multi-route complexity. When multiple reasoning routes are artificially forced, small models accumulate noise rather than useful contrastive signals.
> >
> > It is important to clarify that this multi-route analysis is an ablation strictly for understanding routing complexity. In all main experiments, including TRIDENT, SafetyBench, and Insider Threat, the number of routes is not forced (Please refer to InvThink prompt in Figure 10). Instead, the model is free to determine the number of reasoning paths from the inverse-reasoning prompt. In practice, small models naturally choose a single route, mid-range models (7B–14B) utilize up to five to seven routes, and larger models (32B–72B) benefit from five to seven before saturating, as illustrated in Figure 4. The forced-path ablation exposes this trend, but the operational InvThink pipeline always allows free selection rather than imposing multi-route reasoning on smaller models. We will clarify this distinction in Section 4.2 of the updated manuscript.
> >
> > &nbsp;
> >
> > **References**
> >
> > [1] Li et al. Optimizing Safe and Aligned Language Generation: A Multi-Objective GRPO Approach. 2025.
> >
> > [2] Comanici et al. Gemini 2.5: Pushing the Frontier with Advanced Reasoning, Multimodality, Long Context, and Next Generation Agentic Capabilities. 2025.
> >
> > [3] Shao et al. Your Agent May Misevolve: Emergent Risks in Self-evolving LLM Agents. 2025.
> >
> > [4] Wu et al. TRIDENT: Enhancing Large Language Model Safety with Tri-Dimensional Diversified Red-Teaming Data Synthesis. 2025.
> >
> > [5] Agentic Misalignment: How LLMs could be insider threats https://www.anthropic.com/research/agentic-misalignment

---

### Official Review · Reviewer_6JP3 · 2025-11-01

**Soundness:** 3
**Presentation:** 3
**Contribution:** 3
**Rating:** 6
**Confidence:** 2

**Summary:**

The paper introduces INVTHINK, a safety alignment framework for LLMs that introduces “inverse reasoning” into the generation process. Instead of directly optimizing for safe outputs, INVTHINK trains models to enumerate potential harms, analyze their consequences, then generate safe responses. The process starts with data augmentation, then uses supervised fine tuning, and finally RL.

**Strengths:**

- The paper draws its inspiration from FMEA, making it well rooted in safety engineering work.
- The used pipeline is a standard safety pipeline with some tweaks, which makes it easily adoptable.

**Weaknesses:**

- Requires multi-phase training (SFT + RL), which is resource-intensive, meanwhile a lot of work is aiming for inference-time steering approaches.
- Inverse reasoning adds tokens, potentially increasing inference time.

**Questions:**

- Since evaluation relies on LLM‑as‑judge, how did you control for evaluator bias?
- What is the end‑to‑end training cost and inference overhead ?
- Why is Gemini Pro chosen as a teacher model and how does that impact the performance on the Gemini model?

---

> ### Author Response · Authors · 2025-11-21
> **Rebuttal by Authors [1/2]**
>
> We sincerely thank the reviewer for the time and care devoted to evaluating our submission. We appreciate the acknowledgement that InvThink is well grounded in safety engineering via FMEA and that its pipeline is easy to adopt. Your concerns helped us clarify important aspects of the work, and we have incorporated additional experiments and analysis accordingly.
>
> **W1. Multi-Phase Training vs. Inference-Time Steering Approaches**
>
> We agree with the reviewer that inference-time steering has become increasingly prominent in recent safety research. Importantly, InvThink already supports this paradigm. Our original manuscript included a training-free inference-time prompting version (InvThink, which is different from InvThink SFT and InvThink SFT+RL), evaluated in **Table 2, Figure 2,3,5,6 and 7**. In this configuration, the model is not fine-tuned and instead, it receives the inverse-reasoning instruction at inference time (harm enumeration → consequence analysis → safe output). Despite requiring zero training cost, this inference-only version yields substantial improvements over SafetyPrompt, CoT, and ToT across SafetyBench, TRIDENT, and Insider Threat. For example, on Qwen-3-8B it outperforms SafetyPrompt, CoT, and ToT by **+1–8% absolute** on SafetyBench, reduces TRIDENT risk scores by **15–35%**, and lowers Insider-Threat violations by **3–5×**. Similar trends hold for Gemma-7B-it and Qwen-2.5-7B. These results demonstrate that inverse reasoning is intrinsically effective as an inference-time steering method. We demonstrated further improvement through the SFT and GRPO variants, which are suitable for high-stakes scenarios requiring stronger generalization. Importantly, they do not increase inference-depth or require iterative self-refinement; their inference-time behavior matches that of standard instruction-tuned models. We have updated the manuscript to make this dual-mode design clearer.
>
> **Q1. Evaluation Bias in LLM-as-a-Judge Settings**
>
> We appreciate the reviewer’s concern about evaluator bias and clarify the evaluation pipeline across benchmarks. SafetyBench does not involve subjective LLM scoring. Although we instruct models to respond in a structured format (“Answer: C”), models occasionally deviate. To ensure robustness, we use a lightweight LLM-based extractor only when rule-based parsing fails, and the extracted choice is always fed to a deterministic rule-based evaluation pipeline. At no point does any model judge correctness, so evaluator bias cannot affect SafetyBench results.
>
> For the Insider Threat benchmark, we follow Anthropic’s dual-verification protocol [5]: a response is labeled harmful only when both the LLM judge identifies harmful semantic intent and a regex-based detector identifies explicit harmful traces such as exfiltration or sabotage. This conjunctive structure sharply reduces false positives and negatives caused by judge drift.
>
> To address evaluator bias most thoroughly in TRIDENT, we performed additional multi-LLM ensemble evaluation. All TRIDENT samples were re-evaluated now using Gemini-2.5-Pro, o3-mini, and Claude-3.7-Sonnet, representing three independent model families. Agreement across these judges is high (Pearson 0.819, Spearman 0.831, safety agreement 86.9%, within-1 91.6%). Importantly, InvThink shows the highest cross-judge stability among all methods, with 76.7% exact agreement, 90.4% safety agreement, and the lowest mean deviation (0.319). These trends hold consistently across student model families and across TRIDENT domains. Table 2 now has been updated accordingly in our manuscript.
>
> **W2 & Q2. End-to-End Training Cost and Inference Overhead**
>
> We now provide a detailed breakdown of dataset construction cost, SFT cost, RL cost, and inference-time overhead. The data augmentation step requires generating inverse-reasoning traces with a frontier teacher model; the total cost is modest relative to typical instruction-tuning corpora, and we will include API cost estimates and time required for data generation in the camera-ready version.
>
> Supervised fine-tuning (SFT) for Gemma-7B-it, Qwen-2.5-7B, and Qwen-3-8B requires approximately 27.2 GPU-hours on A40-40GB hardware for three epochs. This is comparable to standard safety-tuning pipelines. Reinforcement learning (RL) is performed using GRPO, which is significantly more lightweight than PPO because it removes the value network component [1]. GRPO training requires approximately 45 GPU-hours on our 4xA40 cluster.
>
> Regarding inference overhead, InvThink supports three operational modes: inference-time prompting only, InvThink-SFT, and InvThink-RL. The inference-only version has zero training cost and introduces ~10–18% additional prompt tokens due to harm enumeration, yielding 25–30% harmfulness reduction on TRIDENT (Figure 8). The SFT and RL versions do not increase inference depth beyond this; unlike iterative self-refinement, InvThink requires only a single forward pass.

---

> ### Author Response · Authors · 2025-11-21
> **Rebuttal by Authors [2/2]**
>
> **Q3. Choice of Gemini-2.5-Pro as Teacher Model and Its Impact**
>
> Gemini-2.5 Pro is used solely as a teacher model to generate inverse-reasoning traces; none of our student models (Gemma-7B-it, Qwen-2.5-7B, Qwen-3-8B) belong to the Gemini family, so there is no overlap in architecture, tokenizer, or pretraining pipeline.
>
> Gemini-2.5 Pro demonstrates state-of-the-art safety performance in cyber-offense prevention, biosecurity reasoning, and misinformation robustness, mentioned in its technical report [2] and in multiple independent safety analyses [3-5] which adopts Gemini-Pro/2.5-Pro as either a teacher or reference judge because of its alignment reliability and calibrated refusal behavior.
>
> In Table 7, we further validates this choice: Gemini-2.5 Pro is the most safety evaluator among the three models. Across all three datasets, Gemini-2.5-Pro provides similar or stricter judgments:
>
> * **SafetyBench:** Gemini-2.5 Pro's 0.85 is **comparable** to Claude 3.7 Sonnet's 0.87 and stronger than o3-mini's 0.83.
>
> * **TRIDENT:** Gemini-2.5 Pro produces the **lowest** score: 1.70, versus 1.75 (Claude) and 1.82 (o3-mini).
>
> * **Insider Threat:** Gemini-2.5 Pro again assigns the **lowest** risk: 0.03, compared with 0.06 (Claude) and 0.09 (o3-mini).
>
> The high inter-judge agreement (Q1) and InvThink’s stability across all evaluators confirm that performance improvements are not tied to any Gemini-specific bias. We will update Appendix A.3 and B to further clarify the separation between teacher and evaluator roles.
>
> &nbsp;
>
> **References**
>
> [1] Shao et al. DeepSeekMath: Pushing the Limits of Mathematical Reasoning in Open Language Models. 2024.
>
> [2] Comanici et al. Gemini 2.5: Pushing the Frontier with Advanced Reasoning, Multimodality, Long Context, and Next Generation Agentic Capabilities. 2025.
>
> [3] Shao et al. Your Agent May Misevolve: Emergent Risks in Self-evolving LLM Agents. 2025.
>
> [4] Wu et al. TRIDENT: Enhancing Large Language Model Safety with Tri-Dimensional Diversified Red-Teaming Data Synthesis. 2025.
>
> [5] Agentic Misalignment: How LLMs could be insider threats https://www.anthropic.com/research/agentic-misalignment

---

### Author Response · Authors · 2025-12-03
**Summary of Contributions and Rebuttal Efforts**

We sincerely thank all reviewers and ACs for their constructive feedback and time. During the rebuttal period, we conducted extensive additional experiments to address every major concern. Below, we summarize the key additions and clarifications.

&nbsp;

### Contributions

**InvThink** introduces inverse reasoning for AI safety, representing a paradigm shift from reactive safety alignment to **proactive harm anticipation before forward reasoning**. Our method instructs models to (1) enumerate potential harms, (2) analyze their consequences, and (3) generate safe outputs constrained by this analysis.

Our key contributions are:

1. **Scaling Properties**: InvThink shows stronger safety improvements with model size compared to existing methods (CoT, ToT, SafetyPrompt), while baseline approaches plateau or degrade (Figure 3).

2. **Safety Tax Mitigation**: Unlike prior safety methods, InvThink preserves or improves general reasoning capabilities (Table 3).

3. **High-Stakes Performance**: Up to 31.8% reduction in harmful responses on TRIDENT and elimination of insider threat behaviors (Table 2).

&nbsp;

### Rebuttal Summary

| Category | Addition | Addresses |
|----------|----------|-----------|
| Multi-LLM Ensemble | Gemini-2.5 Pro, o3-mini, Claude 3.7 Sonnet evaluation (Table 4) | 6JP3 W2 |
| General SFT+RL Baseline | Table 2: InvThink SFT+RL outperforms General SFT+RL across all metrics | QUPy W1 |
| ToT Baseline | Table 2: ToT does not improve safety (Table 2) | JkAD W2 |
| Alternative Teacher Model | gpt-oss-safeguard experiments on SafetyBench, TRIDENT and Insider Threat (Table 2) | 6JP3/QUPy/JkAD |
| WildGuard Reward Model | Consistent gains over General SFT+RL baseline (Table 8) | uDgr W3 |
| Statistical Scaling Analysis | Log-linear regression, dominance analysis | uDgr W1 |
| Training Cost Breakdown | SFT: 27 GPU-hours, RL: 45 GPU-hours; inference overhead: 10-18% tokens; Dataset generation time: 7.8 days | 6JP3/uDgr |
| Optimal Route Statistics | ANOVA p=0.628, 95% plateau at 2-3 routes, CV=3.65% | uDgr Q2 |

&nbsp;

### New Experimental Results

1. **Alternative Teacher Model Experiments**: Trained Qwen-3-8B with gpt-oss-safeguard (instead of Gemini-2.5-Pro) generated dataset:
   - InvThink SFT: SafetyBench 0.82, TRIDENT 1.67, Insider Threat 0.03
   - InvThink SFT+RL: SafetyBench 0.84, TRIDENT 1.43, Insider Threat 0.02
   - Demonstrates InvThink is **teacher-agnostic**, meaning that its benefits stem from structured inverse reasoning rather than teacher-specific knowledge.

2. **General SFT+RL Baseline**: Standard RL also improves performance, but it is not as effective as when it is combined with the InvThink reasoning process.

3. **Tree-of-Thoughts (ToT) Baseline**: ToT does not reliably improve safety, because structured harm-first reasoning matters more than simply generating more thoughts (which is also supported by existing literatures).

4. **Multi-LLM Ensemble Evaluation**: High inter-judge agreement confirms evaluation robustness.

5. **WildGuard Reward Model**: Consistent gains confirm method-agnosticism to reward model choice.

&nbsp;

### Commitments for Camera-Ready

1. **RL data ratio ablation (10%-50%)** : computationally intensive, in progress
Due to the time constraints of the rebuttal process, we will include the ablation experiment in the camera-ready version. As shown below, these experiments require a substantial amount of time to complete.
10% - 21 GPU-hours (estimated) , 4xA40
20% - 45 GPU-hours (measured) , 4xA40
30% - 68 GPU-hours (estimated) , 4xA40
50% - 122 GPU-hours (estimated) , 4xA40
2. **LoRA-based InvThink for larger model**
3. gpt-oss-safeguard experiments extended to **Gemma-7B-it and Qwen-2.5-7B**

&nbsp;

### Final Remarks

1. **Novel Contribution**: InvThink is the first to implement backward harm-first reasoning as a supervised intermediate variable for **LLM safety reasoning**, focusing not on detecting harmfulness but on structurally mitigating it through explicit reasoning steps.

2. **Strong Empirical Results**: InvThink SFT+RL achieves up to 32.0% reduction in harmful responses on TRIDENT, improve safety reasoning in SafetyBench and eliminates insider threat behaviors.

3. **Extensive Validation**: Evaluated across 3 model families (Gemma, Qwen-2.5, Qwen-3), 2 teacher models (Gemini-2.5 Pro, gpt-oss-safeguard), 2 reward models (OpenAI Moderation API, WildGuard), and 3 independent LLM judges (Gemini-2.5 Pro, o3-mini, Claude 3.7 Sonnet).

4. **Teacher-Agnostic**: New experiments with gpt-oss-safeguard directly address the most significant concern raised by multiple reviewers.

5. **Comprehensive Rebuttal**: We addressed every major concern with new experiments, not just explanations. The additions include 6 new experimental results and statistical analyses.

We believe the concerns raised have been thoroughly addressed, and we respectfully ask the AC to consider the substantial improvements made during the rebuttal period.

---

### Note · Program_Chairs · 2026-01-17
**Submission Desk Rejected by Program Chairs**

The following references in this submission do not refer to real documents and/or have major errors in bibliographic information:

 Xiangzhe Lin, Runzhe Yu, Liwen Yuan, Kaitlyn Fan, and Chunting Huang. Rational: Rationalizable and norm-grounded large language models. arXiv preprint arXiv:2503.05021, 2025.